# A Robust Certified Machine Unlearning Method Under Distribution Shift

## Abstract

The Newton method has been widely adopted to achieve certified unlearning. A critical assumption in existing approaches is that the data requested for unlearning are selected i.i.d. (independent and identically distributed). However, the problem of certified unlearning under non-i.i.d. deletions remains largely unexplored. In practice, unlearning requests are inherently biased, leading to non-i.i.d. deletions and causing distribution shifts between the original and retained datasets. In this paper, we show that certified unlearning with the Newton method becomes inefficient and ineffective under non-i.i.d. unlearning sets. We then propose a better certified unlearning approach by performing a distribution-aware certified unlearning framework based on iterative Newton updates constrained by a trust region. Our method provides a closer approximation to the retrained model and yields a tighter pre-run bound on the gradient residual, thereby ensuring efficient $(\epsilon, \delta)$-certified unlearning. To demonstrate its practical effectiveness under distribution shift, we also conduct extensive experiments across multiple evaluation metrics, providing a comprehensive assessment of our approach.

## 1 Introduction

Recent privacy regulations such as the General Data Protection Regulation (GDPR, 2016), the California Consumer Privacy Act (CCPA, 2018), and the Canadian Consumer Privacy Protection Act (CPPA) grant users a *"right to erasure"*, requiring that their personal information and its influence be fully removed upon request. A key requirement of these regulations is the *effectiveness* of data removal. Exact unlearning—retraining a model from scratch without the deleted data—provides the strongest guarantee Xu et al. (2024); Cao & Yang (2015), but is computationally infeasible in practice due to the high frequency and large scale of unlearning requests, as well as the prohibitive cost of repeated retraining. Approximate unlearning methods attempt to remove the influence of specific data without retraining, but they lack rigorous guarantees of correctness Golatkar et al. (2020); Bourtoule et al. (2021); Sekhari et al. (2021); Kurmanji et al. (2023); Brophy & Lowd (2021). This gap has motivated the study of *certified machine unlearning*, where guarantees are provided by bounding the discrepancy between the distributions of the unlearned model and the retrained model, yielding differential-privacy–style assurances Guo et al. (2020).

Certified unlearning has since been applied across multiple domains, including convex linear models Guo et al. (2020); Izzo et al. (2021); Mahadevan & Mathioudakis (2021), non-convex models in computer vision Zhang et al. (2024); Mu & Klabjan (2024); Chowdhury et al. (2024); Basaran et al. (2025), graph neural networks Dong et al. (2024); Wu et al. (2023); Chien et al. (2022), and federated learning models Huynh et al. (2025); Wang & Wang (2025); Wang et al. (2025). However, most existing *post hoc* certified unlearning methods rely on the assumption that the unlearning set is drawn i.i.d. from the same distribution as the original training data. This assumption is unrealistic in practice: at scale, heterogeneous user requests accumulate into systematically biased unlearning sets, inducing *distribution shift* between the original dataset and the retrained dataset. Our key observation is that under such shifts, existing certified unlearning methods inject excessive noise, severely degrading utility of the unlearned model. Although some approaches aim to address this issue Mu & Klabjan (2024); Zou et al. (2025); Liu et al. (2023a); Jia et al. (2024), they either require intrusive pre-processing of the model or fail to provide an upper bound on gradient residual, and thus cannot certify the unlearning.

In this paper, we propose a new certified unlearning method that does not rely on the i.i.d. assumption. Instead, our approach provides a uniform *post hoc* certified unlearning procedure that naturally adapts to distribution shift, yielding tighter bounds and stronger utility guarantees. To the best of our knowledge, this is the first work to systematically identify distribution shift as a fundamental failure mode for certified unlearning and to address it with a distribution-aware framework based on iterative Newton updates constrained by a trust region, which remains effective under non-i.i.d. requests while maintaining competitive efficiency.

## 2 Related Work

**Newton Updates for Certified Machine Unlearning.** Adapting from the local second-order Taylor expansion Polyak (2007); Bui et al. (2024), Newton's method has been widely used to provide certified machine unlearning and is now regarded as the SOTA method across many models. However, two major drawbacks hinder its application to DNNs: reliance on a global convexity assumption and the high computational cost of Hessian estimation. Starting with convex linear models Guo et al. (2020), a line of works has extended Newton-based certified unlearning to DNNs and other non-convex models Koloskova et al. (2025); Dong et al. (2024); Zhang et al. (2024); Liu et al. (2023b) by incorporating regularization to enforce local convexity. Another line of works aims to further broaden applicability by developing more efficient ways to approximate the Hessian within Newton's method Qiao et al. (2024); Dhiman (2019); Ahmed et al. (2025). In this work, we focus on analyzing how Newton-based certified unlearning fails under non-i.i.d. unlearning set construction, where distribution shift between the original and retained datasets undermines its approximation guarantees.

**Evaluation on Unlearning Effectiveness** Evaluation of certified machine unlearning remains Allouah et al. (2024); Wichert & Sikdar (2024) an open challenge, as the effectiveness of unlearning must be reported with precision. In the absence of a universally accepted metric, most prior works rely on membership inference attacks (MIA) Shokri et al. (2017); Hu et al. (2022) to assess unlearning effectiveness Shi et al. (2024); Hayes et al. (2025); Wang et al. (2024); Goel et al. (2022).

## 3 Certified Unlearning under Distribution Shift

**Preliminaries and Notation** Let $\mathbb{D} = \{z_1, z_2, \ldots, z_n\}$ be the training dataset of $n$ samples i.i.d. from the sample space $\mathbb{Z}$. Let $\mathbb{D}_{\text{forget}} \subset \mathbb{D}$ be the forget dataset, and $\mathbb{R} = \mathbb{D} \setminus \mathbb{D}_{\text{forget}}$ the retained dataset. Let $\mathbb{H}$ be the parameter space of a hypothesis class. A learning algorithm $\mathcal{A}$ maps the input training dataset $\mathbb{D}$ to $\mathbb{H}$, producing a parameter vector $w^\star \in \mathbb{H}$ that minimizes the empirical risk:

$$w^\star = \mathcal{A}(\mathbb{D}) = \arg\min_{w \in \mathbb{H}} \mathcal{L}(w, \mathbb{D}), \tag{1}$$

where $\mathcal{L}(\cdot)$ denotes the empirical loss.

Let $\mathcal{U}$ be an algorithm that maps $w^\star$ to $\tilde{w}$, where $\tilde{w}$ denotes the parameters of the unlearned model. To remove the influence of the forgotten data, an approximate unlearning algorithm requires information from the original model $\mathcal{A}(\mathbb{D})$, the full training dataset $\mathbb{D}$, and the retained subset $\mathbb{R}$. We can therefore write

$$\tilde{w} = \mathcal{U}(\mathcal{A}(\mathbb{D}), \mathbb{R}, \mathbb{D}).$$

Alternatively, one can remove the influence of the forgotten data by retraining from scratch on $\mathbb{R}$ using the same learning algorithm $\mathcal{A}$, yielding

$$\widehat{w} = \mathcal{A}(\mathbb{R}).$$

We regard the retrained model $\widehat{w}$ as the *gold standard* for unlearning. Thus, to achieve certified unlearning, the unlearned model $\tilde{w} = \mathcal{U}(\mathcal{A}(\mathbb{D}), \mathbb{R}, \mathbb{D})$ should be designed to be *probabilistically indistinguishable* from the gold standard $\widehat{w} = \mathcal{A}(\mathbb{R})$.

**Certified Unlearning Definition**

**Definition 1.** *For $\delta > 0$, an unlearning algorithm $\mathcal{U}$ satisfies $(\epsilon, \delta)$-certified unlearning if for all $T \subseteq \mathbb{H}$,*

$$\Pr\big[\mathcal{U}(\mathcal{A}(\mathbb{D}), \mathbb{R}, \mathbb{D}) \in T\big] \le e^\epsilon \Pr\big[\mathcal{A}(\mathbb{R}) \in T\big] + \delta, \tag{2}$$

$$\Pr\big[\mathcal{A}(\mathbb{R}) \in T\big] \le e^\epsilon \Pr\big[\mathcal{U}(\mathcal{A}(\mathbb{D}), \mathbb{R}, \mathbb{D}) \in T\big] + \delta. \tag{3}$$

If $\mathcal{A}$ is already $(\epsilon, \delta)$-DP, certified unlearning is trivial. Analogous to DP training, one can perform DP unlearning by bounding $\|\widehat{w} - \tilde{w}\|$ and adding sufficient Gaussian noise. Formally, we introduce the following theorem (Towards Certified Unlearning for Deep Neural Networks) to archive DP certified unlearning in practice:

**Theorem 1.** *Let $\widehat{w}$ be the empirical minimizer over $\mathbb{R}$ and let $\tilde{w} = \text{UNLEARNSTEP}(w^\star, \mathbb{R}, \mathbb{D})$ be a close approximation. If $\|\widehat{w} - \tilde{w}\| \leq \Delta$, where $\Delta$ denotes upper bounded distance between $\tilde{w}$ and $\widehat{w}$, then then:*

$$\mathcal{U}(w^\star, \mathbb{R}, \mathbb{D}) = \text{UNLEARNSTEP}(w^\star, \mathbb{R}, \mathbb{D}) + Y \tag{4}$$

*is $(\epsilon, \delta)$-certified unlearning, where $Y \sim \mathcal{N}(0, \sigma^2 \boldsymbol{I})$ with*

$$\sigma \geq \frac{\Delta\sqrt{2\ln(1.25/\delta)}}{\epsilon}. \tag{5}$$

We argue that our method achieves better certified unlearning under distribution shift by providing a closer approximation to the retrained model, thereby yielding a tighter bound.

## 4 PRIOR WORK AND LIMITATIONS

### 4.1 PRIOR WORK

**Convex Models.** Under the convex setting, the model is often assumed to admit local convexity around the optimum, which allows the Hessian matrix to be computed and used. Consequently, a second-order Newton update serves as an effective approximate unlearning method. In prior work, Guo et al. (2020) proposes a certified unlearning procedure based on a single Newton step applied on linear model. Let $D$ be the training set, $U \subseteq D$ the points to delete, and $R = D \setminus U$ the retained set. The unlearning estimate is

$$\tilde{w} = w^\star - H_D(w^\star)^{-1} \nabla F_R(w^\star),$$

where $H_D(w^\star) = \nabla^2 F_D(w^\star)$ is the Hessian at $w^\star$.

**Non-convex Models.** As proposed in Zhang et al. (2024), the authors address the limitations of applying Newton updates to non-convex models. Specifically, they locally convexify the Newton step by introducing a damping term. By adding a strong regularizer $\lambda I$, the unlearning update becomes

$$\tilde{w} = w^\star - \left(H(w^\star) + \lambda I\right)^{-1} \nabla L(w^\star, \mathbb{R}),$$

where $H(w^\star)$ is the Hessian at $w^\star$ and $\mathbb{R}$ is the retained dataset. The additional term $\lambda I$ shifts the Hessian to be positive-definite and well-conditioned, even when $H(w^\star)$ is indefinite or singular. This damping term is analogous to an $\ell_2$ regularizer, stabilizing the update and ensuring numerical robustness.

### 4.2 PRIOR WORK LIMITATION

However, existing papers with Newton methods update implicitly assume i.i.d sampled unlearned dataset from the underlying training distribution $\mathbb{D} = \{z_i\}_{i=1}^n$ with $z_i \sim_{\text{iid}} P$. In realistic deployments, deletions are *systematically biased*: requests often arrive (i) clustered by user or cohort (e.g., a specific jurisdiction exercising a privacy right) or (ii) concentrated on particular topics or classes (e.g., deletion of sensitive categories). This leads to the over-samples through the distirbution, essentially formed a bias deletion set $U$. Since the retained dataset $\mathbb{R} \subseteq \mathbb{D}$ is also drawn from $P$, removing $U$ induces a non-i.i.d. deletion and consequently skews the distribution of $\mathbb{R}$. This creates a distribution shift as shown below:

**Proposition 1** (Distribution shift from biased deletion). *Let $\mathbb{D} = \{Z_i\}_{i=1}^n \overset{i.i.d.}{\sim} P$, and let $\mathbb{R} = \mathbb{D} \setminus \mathbb{D}_{\text{unlearn}}$ with $m := |\mathbb{D}_{\text{unlearn}}|$ and $r := |\mathbb{R}| = n - m$. If there exists a measurable set $A$ with $P(A) > 0$ such that*

$$\frac{\#(\mathbb{D}_{\text{unlearn}} \cap A)}{\#(\mathbb{D} \cap A)} \neq \frac{m}{n},$$

*then the retained empirical distribution differs from $P$ on $A$, i.e. $\mu_{\mathbb{R}}(A) \neq P(A)$. Consequently, the retained distribution satisfies $P_{\mathbb{R}} \neq P$.*

The proof can be find in Appendix A.1.

As the implicit assumption of no distribution shift breaks down, we demonstrate that certified unlearning methods based on the Newton update collapse: the approximation to the retrained model becomes unreliable, the certified bound on the parameter deviation grows vacuous, and the mechanism is forced to inject overwhelming noise. Taken together, these findings show that prior certified unlearning guarantees are rendered largely ineffective under realistic non-i.i.d. deletions. In particular, we show that:

1. the local second-order Taylor expansion approximation provide poor approximation on retrained model under distribution shift, and

2. the computed upper bound on the residual gradient—that is, the distance between the unlearned model and the retrained model, $\|\tilde{w} - \widehat{w}\|$—is loosely bounded.

in Appendix B, and in Section 4, we demonstrate how our method addresses them.

### 4.3 MOTIVATION

In practice, even modest distribution shifts can substantially loosen the certified upper bound on the gradient residual. This forces excessive noise injection to satisfy $(\epsilon, \delta)$-DP guarantees, degrading utility. As shown in Figure 1, larger shifts—measured by posterior KL divergence—lead to greater accuracy loss in the unlearned model relative to the retrained model. This empirically confirms the limitation of existing methods that we highlight. Our observations indicate a clear increasing trend for the existing method. Section 5 provides a comprehensive study that extends this motivating experiment. These observations motivate us to explore a new unlearning method designed to remain robust under distribution shifts induced by non-i.i.d. deletion sets.

## 5 METHODOLOGY

As formulated in Proposition 4, prior work based on Newton updates fails to achieve efficient certified unlearning under distribution shift. The breakdown arises because (i) the local second-order Taylor expansion provides an inaccurate approximation of the retrained model, and (ii) the residual gradient must be bounded using a global Lipschitz constant, which leads to excessive noise injection and severe utility loss. Our key observation is that non-i.i.d. deletions inflate the *global* gradient and Hessian Lipschitz constants, rendering global Taylor-based bounds vacuous and degrading the accuracy of one-step Newton approximations.

To address this, we propose *TR-certified machine unlearning*, an iterative certified unlearning framework with dynamically adjusted step-size caps, implemented via the trust-region (TR) method. We continue to use Newton updates to remove the influence of the forgotten data, but the TR constraint ensures that each step remains within a region where the local quadratic model is reliable, preventing error blow-up from biased deletions. TR-certified unlearning resolves the core issues of prior methods: (1) by making only local smoothness assumptions, each iteration achieves the same accuracy guarantee as the one-step Newton update in the no-shift case, yielding a closer approximation to the retrained model as the iterates converge; and (2) it enables the formulation of a tight upper bound on residual information, allowing us to calibrate noise without relying on inflated global Lipschitz constants. We next demonstrate how TR-certified unlearning achieves these two goals. To start with the method, we first shows how to define TR problem and subproblem under unlearning setting, we then shows how to bound the collective gradient residual.

**TR Method on Retrain Model Approximation** We approximate the retrained model using a second-order Taylor expansion around the original minimizer $w^\star$:

$$m_t(p) \ = \ f(w_t) \ + \ g_t^\top p \ + \ \tfrac{1}{2}\, p^\top H_{t,\lambda}\, p, \qquad g_t := \nabla f(w_t), \ \ H_{t,\lambda} := \nabla^2 L_{\mathbb{R}}(w_t) + \lambda I, \quad (6)$$

where $p$ denotes the step (or direction) vector from the current point $w_t$. Setting the initialization at the original minimizer $w^\star$, we have the first subproblem:

$$m_0(p) \ = \ f(w^\star) \ + \ g_0^\top p \ + \ \tfrac{1}{2}\, p^\top H_{0,\lambda}\, p, \qquad g_0 := \nabla f(w^\star), \ \ H_{0,\lambda} := \nabla^2 L_{\mathbb{R}}(w^\star) + \lambda I, \quad (7)$$

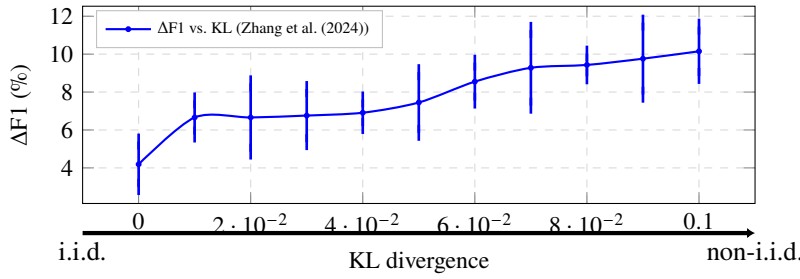

Figure 1: Relationship between KL divergence and $\Delta$F1 for CIFAR-10 with a base CNN model, following the method of Zhang et al. (2024). Vertical error bars show variability across runs. The horizontal arrow highlights the transition from i.i.d. (left) to non-i.i.d. (right).

Notice that solving the first subproblem with the objective function of minimizing loss on the retrained model is mathmatically equivelent to one-step Newton update – which indicates an inaccurate approximation due to large distribution shift. To address this, we adopt a trust-region (TR) method that constrains each Newton update to lie within a trust-region radius. Rather than assuming a global gradient Lipschitz constant, we require only a local Lipschitz condition within the worst-case trust-region ball. Accordingly, the radius must itself be capped to prevent steps from inflating to a global scale. Clipping the maximum trust-region radius is therefore essential to obtain a tight residual-gradient bound and certify unlearning. We begin with the following assumption:

**Assumption 1** (Local smoothness within the trust region). *For each iteration $t$, the per-example loss $\ell(w, x)$ has an $L_t$-Lipschitz continuous gradient in $w$ restricted to the trust region*

$$B(w_t, \bar{\Delta}_t) := \{\, w \in \mathbb{R}^d : \|w - w_t\| \le \bar{\Delta}_t \,\},$$

*i.e.,*

$$\|\nabla\ell(w, x) - \nabla\ell(w', x)\| \le L_t \|w - w'\|, \qquad \forall w, w' \in B(w_t, \bar{\Delta}_t), \ \ x \in \mathbb{R}.$$

*where $B(w_t, \bar{\Delta}_t) := \{\, w \in \mathbb{R}^d : \|w - w_t\| \le \bar{\Delta}_t \,\}$ denotes the trust region around $w_t$ with radius $\bar{\Delta}_t$. We then define the worst-case local Lipschitz constant over $T$ iterations as*

$$L_{\max}(T) := \max_{0 \le t < T} L_t,$$

*which serves as the effective Lipschitz constant in our subsequent upper-bound analysis.*

$L_f(t)$ is a *local* Lipschitz constant it only needs to hold within the trust region $B(w_t, \bar{\Delta}_t)$ rather than across the entire parameter space. In practice, for better implementation, we can estimate it by a few power-iteration steps using Hessian–vector product.

From the Descent Lemma, we obtain a quadratic upper bound ensuring that the true objective value at the next step is no worse than its quadratic approximation. Specifically:

**Proposition 2** (Local Descent Lemma). *Under Assumption 1, for each iteration $t$ the retained objective $f$ satisfies, for any $p$ with $w_t$, $w_t + p \in B(w_t, \bar{\Delta}_t)$,*

$$f(w_t + p) \le f(w_t) + \nabla f(w_t)^\top p + \frac{L_t}{2}\|p\|^2,$$

*where $L_t$ is a Lipschitz constant of $\nabla f$ on $B(w_t, \bar{\Delta}_t)$. For multiple iterations $t = 0, \ldots, T - 1$, the bound holds with*

$$L_{\max}(T) := \max_{0 \le t < T} L_t.$$

The proof is provided in Appendix A.2. This give us the insight to cap the radius bound by L. Hence, steps larger than $\|g_t\|/L$ cannot be justified by the quadratic model unless higher-order terms are tightly controlled, which fail under non-convex setting and limited computation power. We therefore define the clipped trust-region radius with local Lipschitz constant. The effective trust-region radius at step $t$ is given us:

$$\bar{\Delta}_t := \min\{\Delta_t, \ \tfrac{\tau}{L_t}\|g_t\|\}, \qquad g_t := \nabla f(w_t), \ \ \tau \in (0, 1], \tag{8}$$

This ensures that every accepted Newton step remains within a region where the quadratic Taylor expansion is a reliable surrogate. Note that the radius bound is conditioned on the gradient Lipschitz constant rather than the Hessian Lipschitz constant, which yields a tighter control under distribution shift.

Starting from the original minimizer $w_0 = w^\star$, at each iteration $k$ we approximate the retained objective by the quadratic model. Now, the subproblem is formally defined as:

**Definition 2** (TR subproblem). *The trust-region subproblem is then defined as*

$$p_t = \arg\min_{\|p\| \le \bar{\Delta}_t} m_t(p), \qquad w_{k+1} = w_k + p_k, \qquad (9)$$

*where the effective radius is clipped as(using $\tau = 1$)*

$$\bar{\Delta}_t = \min\left\{ \Delta_t, \ \frac{\|g_t\|}{L_t} \right\}, \qquad (10)$$

*with $g_t := \nabla f(w_t)$ and $L_t$ the local Lipschitz constant of $\nabla f$ on $B(w_t, \bar{\Delta}_t)$.*

By appliying Newton update, we can solve the subproblem. In practice, we avoid forming the full Hessian by relying on Hessian–vector products (HVPs) and using truncated conjugate gradient (CG) or LiSSA recursions to compute an approximate step.

We then follow the standard TR procedure to decide whether the step is accepted and to adapt the trust-region radius. Specifically, the model-agreement ratio is defined as

**Definition 3.**

$$\rho_t = \frac{f(w_t) - f(w_t + p_t)}{m_t(0) - m_t(p_t)}. \qquad (11)$$

If $\rho_t \ge \eta_1$ (for some threshold $\eta_1 \in (0, 1)$ we pick), the step is accepted and we set $w_{t+1} = w_t + p_t$; otherwise, the step is rejected and the trust-region radius is contracted. In Appendix C, we show how radius it then updated after solving the TR-subproblem.

We iteratively solve the TR subproblems with approximate Newton updates and adapt the step size until convergence, ensuring that the final accepted step lies within the region governed by the local Lipschitz constant. Rather than relying on a single Newton update, the TR framework defines a region within which the quadratic model is trusted: the Newton direction $-H_t^{-1} g_t$ is preserved as the preferred search direction, but the step size is restricted by a trust-region radius to ensure a valid local gradient smoothness assumption. Consequently, our method achieves a closer approximation to the retained model than a one-step Newton update under distribution shift, while matching the performance of the one-step Newton update when no shift is present. We provide a direct same-metric "closeness" theorem in Appendix A, theorm 3 that formalizes this comparison in terms of the residual gradient.

**Residual Upper Bound Formulation**   We further formulate the residual gradient bound in this section to achieve certified unlearning. With a strong regularizer $\lambda I$, and letting $\|H\|$ denote the spectral norm upper bound on the Hessian of the retained dataset (which we estimate in practice via power iteration), we obtain

$$\|H_{t,\lambda}\| \le \|H_t\| + \lambda, \qquad H_{t,\lambda} := \nabla^2 L_{\mathbb{R}}(w_t) + \lambda I, \qquad (12)$$

where $H_t := \nabla^2 L_{\mathbb{R}}(w_t)$ is the Hessian at the current iterate $w_t$, and $H_{t,\lambda}$ is the damped (regularized) Hessian. Let $w_{t+1} = w_t + p_t$ with $\|p_t\| \le \bar{\Delta}_t$ (the clipped trust-region radius) and define $g_t := \nabla f(w_t)$. Then we can upper bound the gradient of the retained objective at any step $t$ by:

**Proposition 3** (Pre-run gradient bound for TR). *Assume Assumption 1 holds on $B(w_s, \bar{\Delta}_s)$ for $s = 0, \ldots, t - 1$. Then, for any $t \ge 1$,*

$$\|g_t\| \le \|g_0\| + \sum_{s=0}^{t-1} L_s \|p_s\| \le \|g_0\| + L_{\max}(t) \sum_{s=0}^{t-1} \bar{\Delta}_s \le \|g_0\| + L_{\max}(T) \sum_{s=0}^{t-1} \bar{\Delta}_s, \quad (13)$$

*where $\|g_t\| := \|\nabla f(w_t)\|$ and each $L_s$ is a Lipschitz constant of $\nabla f$ on the trust region $B(w_s, \bar{\Delta}_s)$.*

Then, by the Descent Lemma, we obtain a lower bound on the true predicted reduction:

**Theorem 2.** *Let $\varepsilon_{\text{iHVP}}(T, \rho)$ denote the operator-norm error bound on*

$$\left\| \tilde{H}_{T,\lambda}^{-1} - H_\lambda^{-1} \right\|,$$

*derived from the LiSSA analysis. Suppose the local gradient and curvature estimation errors are bounded by $\varepsilon_g$ and $\varepsilon_H$, respectively, where $\varepsilon_H$ captures the error in the Hessian–vector product (HVP). Then, for any iteration $t$, the per-step predicted reduction satisfies*

$$\text{PR}_t \;=\; \max\left\{ 0, \; \underbrace{\text{PR}_t^{\text{est}}}_{\text{model est.}} - \underbrace{\varepsilon_g\, \bar{\Delta}_t}_{\text{grad est.}} - \underbrace{\tfrac{1}{2}\, \varepsilon_H\, \bar{\Delta}_t^2}_{\text{HVP est.}} - \underbrace{\left( \varepsilon_{\text{iHVP}} U_t^2 + (\|H\| + \lambda)\, \bar{\Delta}_t\, \varepsilon_{\text{iHVP}} U_t \right)}_{\text{iHVP step error}} \right\}, \quad (14)$$

*where $U_t := \|g_0\| + L_{\max}(t) \sum_{s=0}^{t-1} \bar{\Delta}_s$ is the pre-run gradient bound.*

**Assumption 2** (Local curvature floor via damping). *Let*

$$f(w) \;=\; L_{\mathbb{R}}(w) + \tfrac{\lambda}{2}\|w\|^2.$$

*We assume there exists $\mu > 0$ such that*

$$\nabla^2 f(w) \;=\; \nabla^2 L_{\mathbb{R}}(w) + \lambda I \succeq \mu I, \qquad \forall\, w \in \mathbb{S},$$

*where $\mathbb{S}$ denotes the region explored by the trust-region iterates $\{w_t\}_{t=0}^T$. Equivalently,*

$$\lambda_{\min}\!\left( \nabla^2 L_{\mathbb{R}}(w) \right) + \lambda \;\geq\; \mu, \qquad \forall\, w \in \mathbb{S}.$$

We formally obtain the approximation bound:

**Corollary 1.** *Given an approximate unlearned model $\tilde{w}$ obtained after $t$ trust-region iterations and the retained minimizer $\widehat{w}$, we can bound their squared distance by*

$$\|\tilde{w} - \widehat{w}\| \;\leq\; \sqrt{\frac{2}{\mu}} \left( 1 - \frac{\eta_1\, \kappa\, \tau\, \mu}{L_{\max}(T)} \right)^{T/2} \sqrt{f(w_0) - f(\widehat{w})}$$

*where $\eta_1 \in (0, 1]$ is the acceptance threshold of the trust-region step, $\kappa \in (0, 1]$ is the solver accuracy factor for approximately solving the TR subproblem, and $\tau \in (0, 1]$ is the clipping factor that scales the effective trust-region radius.*

The proof can be find in Appendix A.3. In practice, if the Hessian matrix remains ill-conditioned and we revert to the Cauchy fallback, then each per-step reduction satisfies a tighter bound. Consequently, we have a tighter approximation bound at the cost of higher computation cost.

## 6   EXPERIMENTS

We design experiments to evaluate the performance of our certified unlearning method under *non-i.i.d.* unlearning sets. We compare against the state-of-the-art certified unlearning methods developed for computer vision models. Our evaluation focuses exclusively on *post-run* certified unlearning methods, since *pre-run* approaches require modifying or pre-processing the model during training and therefore fall outside the scope of this work.

### 6.1   DATASET AND EXPERIMENT SETTINGS

To evaluate the practical effectiveness of our approach, we conduct experiments on MNIST and CIFAR-10. The datasets are split into training/testing with ratios of 6:1 and 5:1, respectively. We use a 3-layer MLP for MNIST and a 3-layer AllCNN for CIFAR-10. To ensure consistency, base models are trained once and reused across all methods. Existing certified unlearning approaches are implemented with their recommended settings, while detailed hyperparameters for TR-certified unlearning are given in Appendix D.

**Non-i.i.d. Sampling Method and Distribution Shift Measurement**   We simulate realistic distribution shifts in unlearning by constructing non-i.i.d. deletion sets that induce systematic mismatches between the original and retrained models. Implementation details are provided in Appendix D.2.

Table 1: F1 and test loss under posterior KL = 0.104. Lower $\Delta$F1 and $\Delta$Loss indicate better utility retention after unlearning.

| | MNIST & MLP | | | CIFAR-10 & AllCNN | | |
|---|---|---|---|---|---|---|
| Method | Retrain (F1/Loss) | Unlearned (F1/Loss) | $\Delta$F1 / $\Delta$Loss | Retrain (F1/Loss) | Unlearned (F1/Loss) | $\Delta$F1 / $\Delta$Loss |
| Guo et al. (2020) | 96.29 / 0.1042 | 92.35 / 0.2184 | 3.94 / 0.1142 | NaN / NaN | NaN / NaN | NaN / NaN |
| Zhang et al. (2024) | 96.77 / 0.1046 | 92.59 / 0.2399 | 4.18 / 0.1353 | 78.43 / 0.6423 | 69.15 / 0.9099 | 9.28 / 0.2676 |
| **TR-Certified (Ours)** | 96.02 / 0.1037 | 94.65 / 0.1811 | **1.37 / 0.0774** | 76.79 / 0.6745 | 73.64 / 0.7925 | **3.15 / 0.1180** |

Table 2: U-MIA accuracy (%) of retrained vs. unlearned models under posterior KL = 0.104. Lower $\Delta$U-MIA indicates less residual leakage.

| | MNIST & MLP | | | CIFAR-10 & AllCNN | | |
|---|---|---|---|---|---|---|
| Method | Retrain (U-MIA) | Unlearned (U-MIA) | $\Delta$U-MIA | Retrain (U-MIA) | Unlearned (U-MIA) | $\Delta$U-MIA |
| Zhang et al. (2024) | 52.14 | 51.25 | 0.89 | 56.00 | 53.27 | 2.73 |
| **TR-Certified (Ours)** | 52.11 | 51.32 | **0.79** | 55.83 | 54.43 | **1.4** |

## 6.2 Evaluation matrix

We evaluate certified unlearning along two major aspects with DP guarantees: effectiveness and utility generalization. Under distribution shift, these require reconsideration, so we redesign the metrics to better reflect certified unlearning performance.

**Utility Generalization**  Under distribution shift, the utility scores on the test set can diverge between the original model $w^*$ and the retrained model $\widehat{w}$. A common practice in unlearning evaluation is to use the difference between the unlearned model $\widetilde{w}$ and the original model $w^*$, i.e., $\Delta_{F1}(\widetilde{w}, w^*)$, as an indicator of the unlearning method's performance. We argue that this comparison is misleading and provides a poor indicator under distribution shift. Instead, we propose that the unlearned model should be evaluated relative to the retrained model, i.e., $\Delta_{F1}(\widetilde{w}, \widehat{w})$, which serves as the appropriate reference point. As discussed in Section 2.1, differential privacy (DP)–based certified unlearning guarantees parameter indistinguishability, $\widetilde{w} \approx_{\epsilon,\delta} \widehat{w}$, which in turn enforces similar utility performance. To measure utility score, we use the micro F1-score of the predictions over the test set $\mathcal{D}_t$ to indicate the utility of the target model. In our results, we report both the difference $\Delta_{F1}(\widetilde{w}, \widehat{w})$ between the F1 scores of the unlearned and retrained models, as well as the absolute F1 score of the unlearned model.

**Effectiveness**  To evaluate the effectiveness of certified unlearning, we conduct U-MIA on both the unlearned model and the retrained model. Membership inference attacks (MIA) aim to determine whether a given data point was used during training by probing the target model. However, under distribution shift they give misleading results: high AUC may reflect either true forgetting of $F$ or spurious drift from distribution shift. Thus, a high AUC score can arise from either successful forgetting of $F$ (the intended effect) or spurious drift and class-mix artifacts introduced by distribution shift. Consequently, classic MIA fails to disentangle true unlearning from distributional effects. To isolate unlearning effectiveness, we instead compare $F$ against truly unseen data ($U$) under the same unlearned model. We adapt the membership inference procedure to distinguish between forget data and genuinely unseen data, following the approach of Hayes et al. (2025). We report the AUC of U-MIA, where values closer to 50% indicate stronger unlearning. Results are reported for both unlearned and retrained models.

## 6.3 Analysis and findings

**Unlearning Utility Study**  In Table 1, we report the KL divergence, F1 score, and loss difference between the state-of-the-art method and ours on the test set. Because the datasets differ in size, we adjust the unlearning ratio to keep the KL divergence between the training sets of the unlearned and retrained models approximately constant (difference < 0.001). To confirm that the unlearning ratio itself does not significantly affect model performance, we provide a separate study in Appendix D.1. Using the retrained model as the baseline, we define $\Delta$F1 = Retrain_F1 − Unlearn_F1. Note that

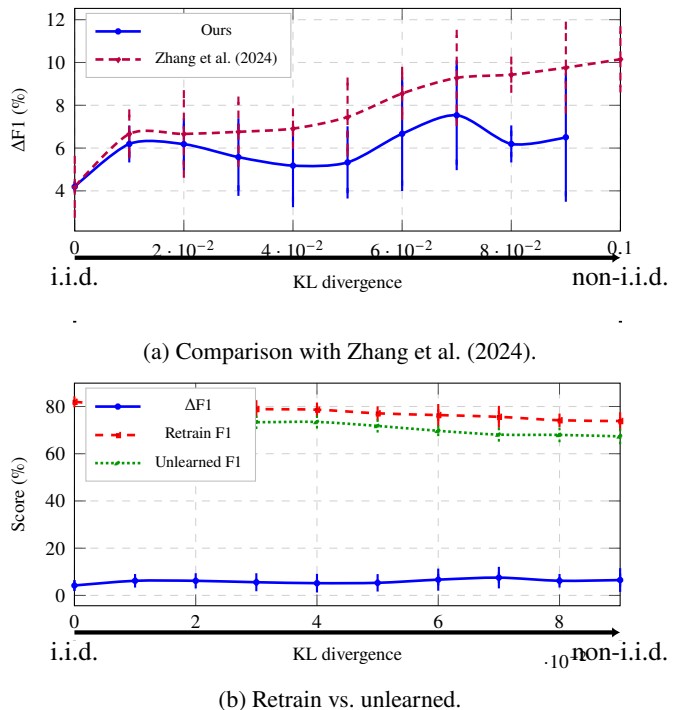

(a) Comparison with Zhang et al. (2024).

(b) Retrain vs. unlearned.

Figure 2: Performance under distribution shift: (a) $\Delta$F1 comparison with Zhang et al. (2024), (b) retrain vs. unlearned F1. Error bars show min–max ranges.

$-\Delta$F1 = Unlearn_F1 $-$ Retrain_F1 is not equivalent to $\Delta$F1. A positive $-\Delta$F1 typically indicates *under-noise*, corresponding to failed certified unlearning.

Under the DP guarantee, a larger $\Delta$F1 on the test set $\mathcal{D}_t$ reflects excessive noise injection. From Table 1, we observe that the state-of-the-art method yields a $\Delta$F1 roughly three times larger than ours, as expected. We also note, however, that the retrained models differ in absolute F1 score and loss, which may partly account for the observed gap. To further validate our findings, Figure 1 evaluates performance under gradually increasing non-i.i.d. deletions, while Figure 2(b) shows that our method consistently outperforms the baseline as KL divergence grows.

**Unlearning Effectiveness Study**   Table 2 compares the U-MIA results for two unlearned models under the same KL setting as Table 1. We apply the single-model U-MIA method to evaluate both the retrained model and the unlearned model separately against the original model. As expected, the retrained model exhibits a larger U-MIA score, since retraining adapts to a different distribution under distribution shift. Similarly, we compute the $\Delta$U-MIA, where a smaller value indicates that the unlearned model more closely aligns with the retrained distribution. We observe that the two methods are nearly identical in U-MIA performance. This outcome is expected, as we deliberately set the Hessian scale to a sufficiently large value such that the global Hessian Lipschitz assumption always holds. This implementation therefore reflects the *best-case* performance of the current state-of-the-art certified unlearning method; in practice, however, performance may degrade if the Hessian scale parameter is underestimated.

Efficiency is another key aspect of certified unlearning, and a detailed efficiency analysis—including on runtime–utility trade-offs is deferred to Appendix E to keep the main text focused on the primary theoretical and empirical results.

## 7   CONCLUSION

Given the limitations of existing certified unlearning methods under i.i.d. deletion sets, we propose a unified framework—TR-Certified Machine Unlearning—that naturally adapts to distribution shift.

We show that TR-Certified Machine Unlearning yields more consistent performance in both utility and effectiveness.

## ETHICS STATEMENT

This work adheres to the ICLR Code of Ethics.[1] We have carefully considered potential ethical concerns related to our study. All datasets used in this paper, CIFAR-10 and MNIST, are publicly available. Membership Inference Attacks (MIA) are employed solely as an evaluation tool to assess the effectiveness of our unlearning method; they are not intended for, nor should they be used in, any other context beyond their designed purpose.

## REPRODUCIBILITY STATEMENT

We are committed to ensuring the reproducibility of our results. Our experimental settings are provided in Section 6, detailed hyperparameters are listed in Table 3, and the proofs of theoretical results are presented in Section 5 and Appendix A.

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

# A   Proposition proofs

*Proof of Proposition 1.* Let $\mu_{\mathbb{R}}$ denote the empirical distribution of the retained dataset $\mathbb{R}$. For any measurable set $A$, we have

$$\mu_{\mathbb{R}}(A) = \frac{\#(\mathbb{R} \cap A)}{|\mathbb{R}|} = \frac{\#\big((\mathbb{D} \setminus \mathbb{D}_{\text{unlearn}}) \cap A\big)}{n - m}.$$

If the deletion set $\mathbb{D}_{\text{unlearn}}$ were sampled uniformly at random, we would have

$$\frac{\#(\mathbb{D}_{\text{unlearn}} \cap A)}{\#(\mathbb{D} \cap A)} = \frac{m}{n},$$

and $\mu_{\mathbb{R}}(A) = P(A)$.

However, by assumption there exists a measurable set $A$ with $P(A) > 0$ such that

$$\frac{\#(\mathbb{D}_{\text{unlearn}} \cap A)}{\#(\mathbb{D} \cap A)} \neq \frac{m}{n}.$$

we then substituting back to the formula for $\mu_{\mathbb{R}}(A)$ then we have: differs from $P(A)$, i.e.

$$\mu_{\mathbb{R}}(A) \neq P(A).$$

This finalize our proof on proposition 1.

*Proof of Proposition 2.* Fix an iteration $t$ and a step $p$ with $w_t, w_t + p \in B(w_t, \bar{\Delta}_t)$. Define the univariate function $\phi(\alpha) := f(w_t + \alpha p)$ for $\alpha \in [0, 1]$. By the chain rule,

$$\phi'(\alpha) = \nabla f(w_t + \alpha p)^\top p.$$

we can then expand to get:

$$f(w_t + p) - f(w_t) = \phi(1) - \phi(0) = \int_0^1 \phi'(\alpha) \, d\alpha = \int_0^1 \nabla f(w_t + \alpha p)^\top p \, d\alpha.$$

Finally after add and then subtract $\nabla f(w_t)$, we can formulate the following inequality:

$$f(w_t + p) - f(w_t) = \nabla f(w_t)^\top p + \int_0^1 \big(\nabla f(w_t + \alpha p) - \nabla f(w_t)\big)^\top p \, d\alpha$$

$$\leq \nabla f(w_t)^\top p + \int_0^1 \big\|\nabla f(w_t + \alpha p) - \nabla f(w_t)\big\| \|p\| \, d\alpha.$$

By Assumption 1, on the ball $B(w_t, \bar{\Delta}_t)$ the gradient is $L_t$–Lipschitz. Since $\|w_t + \alpha p - w_t\| = \alpha\|p\| \leq \|p\| \leq \bar{\Delta}_t$ for $\alpha \in [0, 1]$, we have $\|\nabla f(w_t + \alpha p) - \nabla f(w_t)\| \leq L_t \, \alpha \|p\|$. Therefore,

$$f(w_t + p) - f(w_t) \leq \nabla f(w_t)^\top p + \int_0^1 L_t \, \alpha \|p\|^2 \, d\alpha = \nabla f(w_t)^\top p + \frac{L_t}{2} \|p\|^2.$$

which finalize our proof on proposition 2:

$$f(w_t + p) \leq f(w_t) + \nabla f(w_t)^\top p + \frac{L_t}{2} \|p\|^2.$$

*Proof of Corollary 1.* Let $f(w) = L_{\mathcal{R}}(w) + \frac{\lambda}{2}\|w\|^2$. By Assumption 2 we know that $f$ is $\mu$–strongly convex on the region explored by the TR iterates, hence for the retained minimizer $\widehat{w}$,

$$f(w) - f(\widehat{w}) \geq \frac{\mu}{2} \|w - \widehat{w}\|^2 \qquad \text{and} \qquad \|\nabla f(w)\|^2 \geq 2\mu\big(f(w) - f(\widehat{w})\big). \tag{15}$$

At TR iteration $t$, let $g_t = \nabla f(w_t)$ and $H_{t,\lambda}$ be the (damped) Hessian used in the quadratic model $m_t$. Because the radius is clipped as $\bar{\Delta}_t \leq \frac{\tau}{L_t}\|g_t\|$ (Assumption 1 and Eq. (10)), the Cauchy decrease for the quadratic model gives

$$m_t(0) - m_t(p_t^{\text{Cauchy}}) \geq \frac{\tau}{2L_t} \|g_t\|^2.$$

Let $p_t$ be the (approximately) solved TR step and $\kappa \in (0, 1]$ be the solver accuracy; then

$$m_t(0) - m_t(p_t) \ \geq \ \kappa\big(m_t(0) - m_t(p_t^{\text{Cauchy}})\big) \ \geq \ \kappa\,\frac{\tau}{2L_t}\,\|g_t\|^2.$$

We can further formulate the bound with the loss function f if the step is accepted and we have to combine the agreement ratio: *actual* decrease satisfies

$$f(w_t) - f(w_{t+1}) \ = \ \rho_t\big(m_t(0) - m_t(p_t)\big) \ \geq \ \eta_1\,\kappa\,\frac{\tau}{2L_t}\,\|g_t\|^2.$$

Therefore,

$$f(w_{t+1}) - f(\widehat{w}) \ \leq \ \Big(1 - \frac{\eta_1\kappa\tau\mu}{L_t}\Big)\big(f(w_t) - f(\widehat{w})\big) \ \leq \ \Big(1 - \frac{\eta_1\kappa\tau\mu}{L_{\max}(T)}\Big)\big(f(w_t) - f(\widehat{w})\big),$$

where $L_{\max}(T) = \max_{0 \leq s < T} L_s$.

Finally, we can apply the strong-convexity bound and this gives: equation 15:

$$\frac{\mu}{2}\,\|w_T - \widehat{w}\|^2 \ \leq \ f(w_T) - f(\widehat{w}) \ \leq \ \Big(1 - \frac{\eta_1\kappa\tau\mu}{L_{\max}(T)}\Big)^T \big(f(w_0) - f(\widehat{w})\big),$$

which yields our final results:

$$\|w_T - \widehat{w}\| \ \leq \ \sqrt{\frac{2}{\mu}}\,\Big(1 - \frac{\eta_1\kappa\tau\mu}{L_{\max}(T)}\Big)^{T/2}\sqrt{f(w_0) - f(\widehat{w})}.$$

Setting $\tilde{w} := w_T$ concludes the proof.

**Theorem 3.** *Let $g_t := \nabla f(w_t)$ and $H_t := \nabla^2 f(w_t)$, with $f(w) = L_{\mathcal{R}}(w) + \frac{\lambda}{2}\|w\|^2$.*

*Assume Assumption 1 on $B(w_t, \bar{\Delta}_t)$, $\mu$–strong convexity of $f$ on the region explored by the TR iterates, and a locally $M$–Lipschitz Hessian. For the trust–region step $p_{\text{TR}}$ with radius $r_t = \|g_t\|/\widehat{L}_t$ and $\kappa_{\text{TR}} = \frac{\lambda}{\mu+\lambda}$,*

$$\|\nabla f(w_t + p_{\text{TR}})\| \leq \kappa_{\text{TR}}\|g_t\| + \frac{M}{2}r_t^2.$$

*For the damped Newton step $p_N^{\lambda_N} = -(H_{\text{est}} + \lambda_N I)^{-1}g_t$ with $\alpha_N^{\lambda_N} = \|I - H_t(H_{\text{est}} + \lambda_N I)^{-1}\|$,*

$$\|\nabla f(w_t + p_N^{\lambda_N})\| \leq \alpha_N^{\lambda_N}\|g_t\| + \frac{M}{2}\|(H_{\text{est}} + \lambda_N I)^{-1}\|^2\|g_t\|^2.$$

*As we keep the following,*

$$\text{(C1)} \ \ r_t \leq \|(H_{\text{est}} + \lambda_N I)^{-1}\|\,\|g_t\|, \qquad \text{(C2)} \ \ \kappa_{\text{TR}} \leq \alpha_N^{\lambda_N},$$

*then*

$$\|\nabla f(w_t + p_{\text{TR}})\| \leq \|\nabla f(w_t + p_N^{\lambda_N})\|.$$

*In the no–shift case $H_{\text{est}} = H_t$ and $\lambda_N = \lambda$, the two bounds coincide.*

## B   Existing work limitation extension

**Second-order Taylor expansion approximation.**   The Newton update method can be viewed as a second-order Taylor expansion around the current local minimizer $w^\star$:

$$F_{\mathbb{R}}(w^\star + \Delta) = F_{\mathbb{R}}(w^\star) + \nabla F_{\mathbb{R}}(w^\star)^\top\Delta + \frac{1}{2}\Delta^\top\nabla^2 F_{\mathbb{R}}(w^\star)\,\Delta + R_3(\Delta) \tag{16}$$

The remainder term captures higher-order contributions beyond the quadratic approximation. In prior studies, this remainder is absorbed into the error term and efficiently bounded when computing certified upper bounds on the residual gradient. However, under non-i.i.d. deletion and Proposition 1, a distribution shift arises between the original optimum $w^\star$ and the retrained solution $\widehat{w}$, causing the Taylor remainder to grow and the approximation to deteriorate. Depending on the extent of distribution shift, higher order terms become less accurately approximated.

**Proposition 4** (Taylor remainder under optimizer shift). *Let $F_\mathbb{R}$ have a $\rho$-Lipschitz Hessian in a neighborhood of $w^\star$, i.e. $\|\nabla^2 F_\mathbb{R}(u) - \nabla^2 F_\mathbb{R}(v)\| \le \rho \|u - v\|$. Let $\Delta := \widehat{w} - w^\star$ denote the optimizer shift induced by non-i.i.d. deletion. Then the second-order Taylor expansion at $w^\star$ with remainder terms yields*

$$F_\mathbb{R}(\widehat{w}) = F_\mathbb{R}(w^\star) + \nabla F_\mathbb{R}(w^\star)^\top \Delta + \tfrac{1}{2}\Delta^\top \nabla^2 F_\mathbb{R}(w^\star)\,\Delta + R_3, \qquad |R_3| \le \tfrac{\rho}{6}\|\Delta\|^3, \qquad (17)$$

$$\nabla F_\mathbb{R}(\widehat{w}) = \nabla F_\mathbb{R}(w^\star) + \nabla^2 F_\mathbb{R}(w^\star)\,\Delta + r_2, \qquad \|r_2\| \le \tfrac{\rho}{2}\|\Delta\|^2. \qquad (18)$$

*In particular, since $\nabla F_\mathbb{D}(w^\star) = 0$ but generally $\nabla F_\mathbb{R}(w^\star) \ne 0$ and $\nabla^2 F_\mathbb{R}(w^\star) \ne \nabla^2 F_\mathbb{D}(w^\star)$ under distribution shift, a Newton update constructed at $(F_\mathbb{D}, w^\star)$ incurs the model–mismatch error*

$$e = \nabla F_\mathbb{R}(\widehat{w}) - \left[ \nabla F_\mathbb{R}(w^\star) + \nabla^2 F_\mathbb{R}(w^\star)\Delta \right] = r_2, \qquad \|e\| \le \tfrac{\rho}{2}\|\Delta\|^2.$$

Hence, as the distribution shift and thus $\|\Delta\|$ increases, the higher-order remainder dominates and the local quadratic approximation becomes less accurate. Consequently, using second-order Taylor expansion to approximate the retrained model (and thereby construct the unlearned model) is only valid under local assumptions. Under non-i.i.d. deletion, the displacement error grows quadratically in the optimizer displacement $\|\Delta\|$, causing the quadratic approximation to incur substantial error. At best, this results in overly conservative noise being added, which truncates the utility of the unlearned model.

One natural idea is to consider higher-order Taylor expansions. By incorporating third-order or higher terms, one could in principle obtain a closer approximation to the retrained model, guided by both the Hessian and higher-order derivatives. However, this approach is computationally prohibitive. Forming the full Hessian is infeasible for modern deep neural networks: for a parameter dimension $d$, the Hessian is a $d \times d$ matrix and requires $O(d^2)$ storage and $O(d^2)$–$O(d^3)$ time to compute, depending on whether it is formed explicitly or via matrix–vector products. For large-scale models where $d$ may range from $10^7$ to $10^9$, this easily exceeds memory and runtime limits. Extending to third-order terms is even more impractical: the third-order derivative tensor is of size $d^3$, requiring $O(d^3)$ storage and time.

**Upper bound formulation.** The loose approximation of the retrained model leads to both a large residual gradient and a discrepancy in objective value $F_\mathbb{R}(\tilde{w}) - F_\mathbb{R}(\widehat{w})$, indicating that the unlearned model fails to closely approximate the retrained solution. This implies a larger upper bound $\|\tilde{w} - \widehat{w}\|$, and by Theorem 1, requires injecting more noise to achieve certified unlearning.

For deep neural networks, under the following assumptions: **Assumption 0.1.** The loss function $\ell(w, x)$ has an $L$-Lipschitz gradient with respect to $w$. **Assumption 0.2.** The loss function $\ell(w, x)$ has an $M$-Lipschitz Hessian with respect to $w$. We can bound the reminder term of second-order Taylor expansion with the following(quoted from the paper Towards Ceritifed Unlearning in DNN):

**Proposition 5.** *The second-order Taylor expansion error satisfies*

$$\|\nabla F_\mathbb{R}(\tilde{w})\| \ \le \ \tfrac{M}{2}\|\widehat{w} - w^\star\|^2,$$

so bounding it requires $M$ to be valid on the entire path between $w^\star$ and $\widehat{w}$. Under distribution shift, $\|\widehat{w} - w^\star\|$ is large, and local curvature can vary sharply. Thus only a large global $M$ makes the bound valid.

In the existing work *Towards Certified Unlearning for Deep Neural Networks* derives the upper bound

$$\|\tilde{w} - \widehat{w}\|_2 \ \le \ \frac{2C(MC + \lambda) + G}{\lambda + \lambda_{\min}} + \frac{16\ln d/\rho}{\lambda + \lambda_{\min}} + \frac{1}{16}(2LC + G).$$

Substituting such values of $M$ and $L$ into the inequality yields an extremely loose upper bound. Since noise variance is calibrated to this bound, the certified mechanism adds excessive noise, degrading utility.

A possible remedy is to instead assume local Lipschitz constants. Yet, in the non-i.i.d. unlearning setting, deletions cause sharp changes in the Hessian, causing local Lipschitz constants still be unreasonably large, leaving the bounds practically ineffective.

Thus, prior certified unlearning approaches based on Newton updates break down under distribution shift: due to inaccurate approximations and overly loose bounds, they are guaranteed to collapse into excessive noise injection, rendering the unlearned model inefficient and ineffective in practice. For certain deep models, this issue can arise more easily and cause greater degradation of utility due to the model's sensitivity to biased deletions. In particular, graph DNNs are especially vulnerable, since biased unlearning datasets can naturally emerge from the graph structure, amplifying distribution shifts and worsening the breakdown of certified unlearning guarantees.degradation of utility. In next section we propose a new certified unlearning method that provides a better approximation on retrained model through iterative steps and provides a tighter bound.

## C   TR STEP UPDATE

In each iteration, the radius is updated after solving the TR-subproblem with the following:

$$\Delta_{t+1} = \begin{cases} \gamma_{\text{inc}} \, \Delta_t, & \rho_t \geq \eta_2, \\ \Delta_t, & \eta_1 \leq \rho_t < \eta_2, \\ \gamma_{\text{dec}} \, \Delta_t, & \rho_t < \eta_1, \end{cases} \tag{19}$$

where $0 < \gamma_{\text{dec}} < 1 < \gamma_{\text{inc}}$ are contraction and expansion factors. This mechanism ensures that updates remain within regions where the quadratic model is reliable, preventing over-approximation under distribution shift.

We estimate the initial curvature bound $\widehat{L}_0$ via a few matrix-free Hessian–vector power iterations, and set

$$r_t = \frac{\|g_t\|}{\widehat{L}_t}.$$

In implementation of our method, we avoid recalculating $\widehat{L}_t$ from scratch each time by using an iterative update

$$\widehat{L}_t = \alpha_L \, \widehat{L}_{t-1}, \qquad \alpha_L \geq 1,$$

which provides a fast, conservative bound for the next step.

## D   UNLEARNING PARAMETER

Table 3 demonstrates our parameters on CIFAR10 and MNIST implementation.

| Hyperparameter | CIFAR-10 | MNIST |
|---|---|---|
| Weight decay | 0.0005 | 0.0005 |
| Learning rate ($\eta$) | 0.001 | 0.001 |
| $C$ | 20 | 10 |
| Cert. weight decay | 20 | 1 |
| Number unlearned samples | 8000 | 8000 |
| Trust-region iterations $T$ | 10 | 5 |
| Initial radius $\Delta_0$ | 1.0 | 1.0 |
| Acceptance threshold $\eta_1$ | 0.1 | 0.1 |
| Expansion threshold $\eta_2$ | 0.9 | 0.9 |
| Radius contraction $\gamma_{\text{dec}}$ | 0.5 | 0.5 |
| Radius expansion $\gamma_{\text{inc}}$ | 2.0 | 2.0 |
| $\tau$ (clipping factor) | 1.0 | 0.8 |
| $\lambda$ bump | Disabled | Disabled |

Table 3: Hyperparameter settings for CIFAR-10 and MNIST experiments.

Table 4: KL divergence values versus unlearning ratio for different certified unlearning methods.

(a) Towards Certified Unlearning for DNNs Zhang et al. (2024)

| KL Divergence | Unlearning Ratio |
|---|---|
| 0.000001 | 1000 |
| 0.000003 | 2000 |
| 0.000006 | 3000 |
| 0.000010 | 4000 |
| 0.000007 | 5000 |
| 0.000012 | 6000 |
| 0.000008 | 7000 |
| 0.000015 | 8000 |
| 0.000010 | 9000 |
| 0.000018 | 10000 |

(b) TR-Certified Unlearning (Ours)

| KL Divergence | Unlearning Ratio |
|---|---|
| 0.000004 | 1000 |
| 0.000005 | 2000 |
| 0.000006 | 3000 |
| 0.000011 | 4000 |
| 0.000007 | 5000 |
| 0.000012 | 6000 |
| 0.000009 | 7000 |
| 0.000013 | 8000 |
| 0.000010 | 9000 |
| 0.000017 | 10000 |

### D.1 UNLEARNING RATIO AND KL STUDY

We conduct a third experiment on both methods to clarify how the $\Delta F1$ score changes as the unlearning ratio increases up to 10,000 samples (the maximum used in our CIFAR-10 experiments). The results are reported in Appendix D.2. Across the entire range, the maximum difference is less than 0.00001%, verifying our observation that the unlearning utility of both methods is not significantly influenced by the unlearning ratio. Table 4 reports the KL divergence as different unlearning ratios are applied to two models on the CIFAR-10 dataset under an i.i.d. unlearning set. We observe that increasing the unlearning ratio only changes the overall distribution by approximately $10^{-6}$, which is negligible compared to the distribution shifts induced by non-i.i.d. unlearning sets, typically on the order of $10^{-2}$.

## E UTILITY–EFFICIENCY TRADE-OFF ANALYSIS

We evaluate the trade-off between utility retention and computational efficiency by separating the analysis of predictive performance from running time. Table 5 reports

$$\Delta F_1 = \left| F_1(w_{\text{retrain}}) - F_1(w_{\text{unlearn}}) \right|$$

between the unlearned and retrained models at a fixed posterior KL constraint of 0.104, where a smaller $\Delta F_1$ indicates closer utility to retraining. Table 6 summarizes the corresponding wall-clock unlearning time across datasets and architectures.

With the results in Tables 5 and 6, the performance of our method is best viewed holistically. Our approach consistently outperforms the baselines by roughly 3× on the evaluated utility metric, at the cost of higher unlearning time. This runtime is largely governed by the maximum iteration budget rather than instability or divergence. To further understand this effect, Table 7 reports unlearning time on CIFAR-10 with AllCNN under different target KL thresholds. When the target KL is below 0.10, our method's runtime remains stable (around 92–96 seconds) and is reduced by nearly half compared to the case of KL = 0.10.

There is an inherent trade-off between runtime and unlearning performance; within this trade-off, our method—though slower than the baselines yet still faster than retraining—achieves substantially higher $F_1$, yielding a favorable utility–efficiency balance.

## F THE USE OF LARGE LANGUAGE MODELS

We used Large Language Models solely to correct minor grammatical issues in the writing. Specifically, in Sections 1 and 2, we employed GPT-5 for grammar refinement. Large Language Models made no contribution to any other part of the paper.

Table 5: $\Delta F_1$ across datasets and models at KL = 0.104. Lower $\Delta F_1$ indicates better utility closeness to retraining.

| Method | MNIST & MLP $\Delta F_1$ | CIFAR-10 & AllCNN $\Delta F_1$ | SVHN & ResNet-18 $\Delta F_1$ |
|---|---|---|---|
| Guo et al. (2020) | 3.94 | NaN | NaN |
| Zhang et al. (2024) | 4.18 | 9.28 | 11.20 |
| **TR-Certified (Ours)** | **1.37** | **3.15** | **4.09** |

Table 6: Running time (wall-clock seconds) across datasets and models at KL = 0.104.

| Method | MNIST & MLP (s) | CIFAR-10 & AllCNN (s) | SVHN & ResNet-18 (s) |
|---|---|---|---|
| Guo et al. (2020) | 28.80 | NaN | NaN |
| Zhang et al. (2024) | 14.03 | 47.70 | 105.13 |
| **TR-Certified (Ours)** | 48.23 | 158.20 | 354.72 |

## F.1 Non i.i.d Sampling

For MNIST and CIFAR-10 (each containing 10 balanced classes), we introduce a *bias coefficient* $b_c$ for each class $c$. A deletion set is then sampled proportionally to $b_c$, so that classes with larger coefficients are more likely to be forgotten. Random assignment of coefficients produces heterogeneous shifts across runs, which complicates controlled comparisons. To solve this, we fix a bias pattern on selected classes (e.g., {0 : 99, 7 : 99}) and vary the unlearning ratio to induce controlled levels of distribution shift. This design ensures that experiments with the same shift size exhibit comparable distribution distances, enabling reproducible and systematic evaluation of certified unlearning under biased unlearning sets.

Table 7: Running time (wall-clock seconds) on CIFAR-10 with AllCNN under different target KL thresholds.

| Method | **KL** = 0.02 | **KL** = 0.04 | **KL** = 0.06 | **KL** = 0.08 | **KL** = 0.10 |
|---|---|---|---|---|---|
| Guo et al. (2020) | NaN | NaN | NaN | NaN | NaN |
| Zhang et al. (2024) | 43.55 | 43.20 | 41.80 | 43.08 | 47.70 |
| **TR-Certified (Ours)** | 95.88 | 95.45 | 96.26 | 92.01 | 159.20 |

