# OpenReview forum: "A Robust Certified Machine Unlearning Method Under Distribution Shift"
_ICLR.cc/2026/Conference — Submitted to ICLR 2026_

### Official Review · Reviewer_fafB · 2025-10-31

**Soundness:** 4
**Presentation:** 3
**Contribution:** 3
**Rating:** 4
**Confidence:** 4

**Summary:**

This paper explores exact certified unlearning for non-linear neural networks under the non-IID retain data assumption. The authors point out that, in this setting, a single-step Newton update with added noise is inefficient. To address this, they propose performing multiple Newton updates within a trust region (TR) framework, demonstrating that this approach yields an unlearned model that more closely approximates the retrained model.

**Strengths:**

1. The paper is well written and easy to follow.

2. The problem statement is novel — the authors address the non-IID data deletion scenario, which is often overlooked in existing certified unlearning literature.

3. Extensive theoretical rigors are demostrated, providing proofs and analyses to support the proposed method.

**Weaknesses:**

1. Although the paper provides extensive theoretical analysis, the experimental evaluation is limited. Only a single baseline is compared, whereas other exact unlearning methods based on Newton updates should have been included for a more comprehensive comparison.

2. In addition, the experiments are conducted solely on plain MLP architectures, which limits the practical relevance of the results — evaluation on more complex or modern architectures (e.g., CNNs or Transformers) would strengthen the claims.

3. Although the paper claims that a single-step Hessian update is unreliable for non-IID retain sets, I did not find a formal proof supporting this statement. Moreover, an explicit example or illustration of what constitutes a non-IID retain set and when such a scenario arises would greatly help readers understand the motivation. I could not locate this explanation in the experiments section either—perhaps I overlooked it. Since the entire premise of the paper relies on this assumption, providing a formal justification and a concrete example would significantly strengthen the work.

**Questions:**

1. How is the local Lipschitz constant computed at each step to solve the optimization problem? Additionally, is it practically feasible to estimate this constant for large-scale neural networks, where computing exact Hessians or higher-order terms can be computationally prohibitive?

2. What is the effect of the forget data size on the performance of the proposed method? Specifically, how does increasing or decreasing the number of samples to be unlearned influence the convergence, stability, or closeness of the unlearned model to the retrained model?

---

> ### Author Response · Authors · 2025-11-21
> **Response to Reviewer fafB (Part 1)**
>
> We sincerely thank the reviewer for their careful reading and for the thoughtful effort to help improve our paper. We are glad that the reviewer is interested in our work. We also truly appreciate the recognition of our work’s in terms of soundness, presentation and contribtuion. We have addressed all concerns in detail and kindly invite the reviewer to re-evaluate our work based on our responses
>
> > **W1: Although the paper provides extensive theoretical analysis, the experimental evaluation is limited. Only a single baseline is compared, whereas other exact unlearning methods based on Newton updates should have been included for a more comprehensive comparison.**
>
> **A1:** We appreciate the reviewer's insight to provide more baselines[1]. In response to your concern, we have conducted an additional experiment on another baseline that also utilizes Newton updates for certified unlearning. We are currently working on incorporating more relevant baselines into our evaluation and will update the paper accordingly.
> **Table 1.** $ \Delta \mathrm{F}_1 $ across datasets and models at KL = 0.104.
> | Method                  | MNIST & MLP $ \Delta \mathrm{F}_1 $ | CIFAR-10 & AllCNN $ \Delta \mathrm{F}_1 $ | SVHN & ResNet18 $ \Delta \mathrm{F}_1 $ |
> |-------------------------|------------------|------------------------|---------------------|
> | Guo et al. (2020)       | 3.94               | NaN                    | NaN                  |
> | Zhang et al. (2024)     | 4.18             | 9.28                   | 11.20                 |
> | **TR-Certified (Ours)** | **1.37**         | **3.15**               | **4.09**                 |
>
> It is clearly that for our proposed method outperform all the baselines.
>
> > **W2: In addition, the experiments are conducted solely on plain MLP architectures, which limits the practical relevance of the results — evaluation on more complex or modern architectures (e.g., CNNs or Transformers) would strengthen the claims.**
>
> **A2:** We thank the reviewer for raising this important point regarding model architecture diversity. We would like to clarify that we have already conducted experiments on non-convex model architectures, as reported in Table 1 and Table 2 of our paper. Specifically, we evaluated our method on a CNN architecture (AllCNN). To further address the reviewer's concern, we have run additional experiments on a ResNet architecture (ResNet18). The following Table 2 presents our experimental results using the newly added model architecture. Table 1 reports $ \Delta \mathrm{F}_1 $ between the unlearend model and the retrained model. The lower the $ \Delta \mathrm{F}_1 $, the better the unlearning method.
>
> **Table 2.** $ \Delta \mathrm{F}_1 $ across datasets and models at KL = 0.104.
> | Method                  | MNIST & MLP $ \Delta \mathrm{F}_1 $ | CIFAR-10 & AllCNN $ \Delta \mathrm{F}_1 $ | SVHN & ResNet18 $ \Delta \mathrm{F}_1 $ |
> |-------------------------|------------------|------------------------|---------------------|
> | Guo et al. (2020)       | 3.94               | NaN                    | NaN                  |
> | Zhang et al. (2024)     | 4.18             | 9.28                   | 11.20                 |
> | **TR-Certified (Ours)** | **1.37**         | **3.15**               | **4.09**                 |
>
> We observe from Table 1 that our method continues to outperform other baseline methods across all evaluated architectures, including the newly added ResNet18. This provides additional empirical evidence that our approach performs consistently on more complex and modern architectures, underscoring the practical relevance and robustness of our method and evaluation.

---

> ### Author Response · Authors · 2025-11-21
> **Response to Reviewer fafB (Part 2)**
>
> > **W3: Although the paper claims that a single-step Hessian update is unreliable for non-IID retain sets, I did not find a formal proof supporting this statement. Moreover, an explicit example or illustration of what constitutes a non-IID retain set and when such a scenario arises would greatly help readers understand the motivation. I could not locate this explanation in the experiments section either—perhaps I overlooked it. Since the entire premise of the paper relies on this assumption, providing a formal justification and a concrete example would significantly strengthen the work.**
>
> **A3:** We thank the reviewer for pointing this important aspects of our work. We would like to clarify that we already provide both a formal analysis and concrete examples for the unreliability of single-step Hessian updates under non-IID retain sets.
>
> In appendix section B (Proposition 4, Taylor remainder under optimizer shift), we systematically analyze the mathematical reasons behind the failure of previous single-step Hessian updates on non-i.i.d.\ forget sets. Specifically, we formally shows that when non-i.i.d.\ deletions induce an optimizer displacement $\Delta$, and the Taylor remainder grow as $\mathcal{O}(\|\Delta\|^2)$, making the single-step Newton update unreliable under non-i.i.d.\ setting.
>
> Intuitively, for vision models trained on CIFAR-10, user deletion requests are often class-biased. In extreme case for example, all unlearn request will target to a single class (for example the class dog). In our paper, we did instantiate exactly this class-biased non-IID sampling scheme in Appendix E.1 and use it throughout our experiments, where we empirically observe the breakdown of the single-step Newton baseline.
>
> > **Q1.1: How is the local Lipschitz constant computed at each step to solve the optimization problem?**
>
> **A4.1:**  We thank the reviewer for raising this question. At iteration $t$, we maintain an independent upper bound $\widehat L_t$ on the local gradient–Lipschitz constant. We initialize $\widehat L_t$ via Hessian–vector power iterations. In implementation, we avoid calculating the $\widehat L_t$ every time, so that we apply an iterative update which is defined as:
>
> $$
> \widehat L_t = \alpha_L\,\widehat L_{t-1}, \qquad \alpha_L \ge 1,
> $$
>
> to quickly get the $\widehat L_t$ for next step. We omit this part in the paper as this is only an implementation technique we used. In respond, we will add this clarification in appendix.
>
> > **Q1.2: Additionally, is it practically feasible to estimate this constant for large-scale neural networks, where computing exact Hessians or higher-order terms can be computationally prohibitive?**
>
> **A4.2:**  We thank the reviewer for raising this question.
> For larger neural network, this implementation details on iterative update is important as it avoids computational heavy calculation from approximating Hessian matrix. We believe this is an important future direction, and we plan to verify on our performance on large models and explore how to best extend our methods to large models. We appreciate the reviewer for highlighting this valuable question.

---

> ### Author Response · Authors · 2025-11-21
> **Response to Reviewer fafB (Part 3)**
>
> > **Q2: What is the effect of the forget data size on the performance of the proposed method? Specifically, how does increasing or decreasing the number of samples to be unlearned influence the convergence, stability, or closeness of the unlearned model to the retrained model?**
>
> **A5:** We thank the reviewer for raising this question. We have already examined the effect of the number of forget samples on the KL divergence between the original and retrained models. Appendix E (Table 4) reports how this KL divergence changes as we increase the size of the forget set.
>
> It is insightful to consider the influence of forget data sizes on the proposed method given a non IID unlearning set. In response, we further analyze the influence of unlearning ratio on convergence and stability of our proposed method by running an addition set of experiments. Specifically, we fix the iterative steps and other hyper-parameters and record the time to reflect the convergence and stability performance. We have shown in our paper that closeness is bounded and it is not relevant with unlearning ratio.
>
> In Table 3, we report the time in seconds required to perform our unlearning method on the CIFAR-10 dataset using the AllCNN architecture. Since all other hyperparameters are fixed, longer runtime directly indicates slower convergence or instability of the unlearning procedure.
>
> **Table3: Running time on CIFAR-10 with AllCNN at different unlearning ratios**
>
> | **Method**              | **Unlearning = 1000** | **Unlearning = 3000** | **Unlearning = 5000** | **Unlearning = 7000** |
> |-------------------------|-----------------------:|-----------------------:|-----------------------:|-----------------------:|
> | **TR-Certified (Ours)** | 94.24                  | 93.21                  | 98.37                  | 94.80                  |
>
> As shown in Table 3, the unlearning time remains consistently around 94 seconds across a wide range of unlearning sample sizes. This indicates that our method exhibits stable and reliable convergence behavior, even as the unlearning ratio varies substantially.
>
> Once again, we sincerely appreciate your detailed comments on our paper. Please share any further concerns with us.
>
> [1]Guo, Chuan, et al. "Certified data removal from machine learning models." arXiv preprint arXiv:1911.03030 (2019).

---

> ### Author Response · Authors · 2025-11-27
>
> Thank you again for your time and effort in reviewing our work.
>
> We hope you are enjoying the Thanksgiving holiday, and we hope you have had an opportunity to look over our responses. We appreciate your recognition and positive feedback on our work. We believe we have addressed the concerns raised about our paper, and we kindly hope the reviewer will consider updating their score accordingly.
>
> As the discussion period will end in less than a week, we wish to leave enough time to address your futher concerns. If you have any further questions, we are happy to engage on addressing them.

---

### Official Review · Reviewer_t1Q3 · 2025-10-31

**Soundness:** 3
**Presentation:** 3
**Contribution:** 2
**Rating:** 6
**Confidence:** 4

**Summary:**

This paper studies machine unlearning problem and proposes to use trust region to perform certified machine unlearning with non-i.i.d. data deletion. Instead of one-step Newton, the paper proposes to iteratively perform update to keep track of data shifts. The method is then validated by numerical studies.

**Strengths:**

The machine unlearning problem is timely. Being able to handle non iid data removal is important. The paper is generally clearly written. By using iterative updates, the method only rely on local Lipschitz constants. The use of trust region method make it possible to accommodate non-iid data removal and distribution shift.

**Weaknesses:**

The novelty is somewhat limited, this paper combines a few existing ideas. More importantly, unlearning is usually from the context of multi-agent setup. The usage of TR would make it very difficult to generalize to the scenarios of federated learning setting, since coordinated global search required by TR is either very expensive or not possible to implement in distributed setting. The authors need to really motivate well the usage of centralized unlearning methods in any practical settings.

The proposed TR method is iterative, requiring multiple steps, whereas the baseline is a single-step update. The TR method is inherently more computationally expensive. The paper completely omits any analysis of runtime or computational overhead, making it impossible to assess the trade-off between the claimed utility improvements and efficiency.

**Questions:**

Can the authors compare their method against Qiao or other works that efficiently approximates Newton step? While it is believable that those method would fail under non iid setting, there is no convincing evidence to show improvement. It would be good to add that comparison.

---

> ### Author Response · Authors · 2025-11-21
> **Response to Reviewer t1Q3 (Part 1 -1/2)**
>
> We sincerely thank the reviewer for their careful reading and thoughtful efforts to help improve our paper. We are grateful for the reviewer’s recognition of the strengths of our work. Below, we provide our point-by-point responses.
>
> > **W1: The novelty is somewhat limited, this paper combines a few existing ideas.**
>
> **A1.1:** We appreciate the reviewer’s perspective regarding the perceived novelty and would like to provide additional clarification on this point. We would like to clarify and reiterate that the novelty of our method is twofold. First, we provide a formal definition of the distribution shift induced by unlearning, which is a challenge that prior certified unlearning works have largely taken as default and left unexplored.
>
> Second, building on this formulation, we derive residual gradient bounds and propose, to the best of our knowledge, the first practical certified unlearning framework that explicitly and effectively perform certified unlearning under distribution shift. Specifically, the difficulty lies in deriving a tight upper bound on the residual gradient within the trust-region framework, rather than directly combining framework. We believe this contribution directly strengthens the certified unlearning literature and highlights the novelty of our method, as no existing work has formally defined or tackled this problem.
>
> > **W1.2: More importantly, unlearning is usually from the context of multi-agent setup. The usage of TR would make it very difficult to generalize to the scenarios of federated learning setting, since coordinated global search required by TR is either very expensive or not possible to implement in distributed setting. The authors need to really motivate well the usage of centralized unlearning methods in any practical settings.**
>
> **A1.2:** We would like to emphasize that centralized deployment is a primary and practically important setting for machine unlearning. A recent survey on machine unlearning recognizes centralized unlearning as the main domain and foundation for most existing methods [1]. Given this context, we argue that our method remains strongly practically relevant. For example, consider the following realistic centralized deployment scenarios:
>
> (a) single-controller model owners (cloud/SaaS providers or on-prem enterprises) that already retrain centrally on first-party data.
>
> (b)regulated deletion requests (DSAR/RTBF) handled by a data controller that must produce a single auditable model artifact.
>
> (c) shared models served behind an API, where tenants do not train locally but the provider is obligated to delete/forget centrally.
>
> Additionally, we points out that implementing our proposed method to federated learning setting is practical. Given that our method operates through post-training weight editing, we argue that our method demonstrates stronger generalizability than existing certified unlearning approaches such as SISA [2] and Rewind [3], both of which rely on specific pre-training strategies.
>
> We agree this is an important future direction that we can explore in our future work. We will investigate how to integrate our method into non-centralized learning framework. We appreciate the reviewer for highlighting this valuable question.

---

> ### Author Response · Authors · 2025-11-21
> **Response to Reviewer t1Q3 (Part 2 -2/2)**
>
> > **W2: The proposed TR method is iterative, requiring multiple steps, whereas the baseline is a single-step update. The TR method is inherently more computationally expensive. The paper completely omits any analysis of runtime or computational overhead, making it impossible to assess the trade-off between the claimed utility improvements and efficiency.**
>
> **Q2:** We thank the reviewer for highlighting the importance of evaluating computational efficiency. In response, we have conducted additional experiments to measure the runtime of our proposed method compared to the other baseline methods. To better visulize the results, we seperate the utility evaluatioin from the time evaluation. Table 1 reports the $ \Delta \mathrm{F}_1 $ between unlearend model and the retrained model. Specifically, $ \left| \Delta\mathrm{F} _ 1\left(w _ {\text{retrain}}\right) - \Delta\mathrm{F} _ 1\left(w _ {\text{unlearn}}\right) \right| $ . A smaller $ \Delta \mathrm{F}_1 $ indicates better utlity closeness. We refer to Section 6. for evaluation details.
>
> Table 2 demonstrates the time recorded in seconds on different datasets and model architecture during unlearning.
>
> **Table 1.** $ \Delta \mathrm{F}_1 $ across datasets and models at KL = 0.104.
> | Method                  | MNIST & MLP $ \Delta \mathrm{F}_1 $ | CIFAR-10 & AllCNN $ \Delta \mathrm{F}_1 $ | SVHN & ResNet18 $ \Delta \mathrm{F}_1 $ |
> |-------------------------|------------------|------------------------|---------------------|
> | Guo et al. (2020)       | 3.94               | NaN                    | NaN                  |
> | Zhang et al. (2024)     | 4.18             | 9.28                   | 11.20                 |
> | **TR-Certified (Ours)** | **1.37**         | **3.15**               | **4.09**                 |
>
> **Table 2.** Running time (wall-clock seconds) across datasets and models at KL = 0.104.
> | **Method**               | **MNIST & MLP (s)** | **CIFAR-10 & AllCNN (s)** | **SVHN & ResNet-18 (s)** |
> |--------------------------|---------------------|----------------------------|---------------------------|
> | Guo et al. (2020)        | 28.80               | NaN                        | NaN
> | Zhang et al. (2024)      | 14.03               | 47.7                       | 105.13
> | **TR-Certified (Ours)**  | 48.23               | 158.2                      | 354.72
>
> With the results in Table 1 and Table 2, we argue that the performance of our method should be evaluated holistically. We observe that our method has consistently outperform other baselines by approximately 3× on the evaluated metrics. As a trade-off, our method's unlearning time  is longer than other baselines. However, this runtime is largely influenced by the maximum iteration budget rather than by instability or divergence. To quantify this, we conducted another experiment summarized in Table 3. Table 3 reocrds unlearning time in seconds between different method on CIFAR-10 with AllCNN under different target KL thresholds.
>
> Table 3 **Table: Running time (wall-clock seconds) on CIFAR-10 with AllCNN at different KL thresholds**
>
> | **Method**               | **KL = 0.02** | **KL = 0.04** | **KL = 0.06** | **KL = 0.08** | **KL = 0.10** |
> |--------------------------|--------------:|--------------:|--------------:|--------------:|--------------:|
> | Guo et al. (2020)        | NaN           | NaN           | NaN           | NaN           | NaN           |
> | Zhang et al. (2024)      | 43.55         | 43.20         | 41.80         | 43.08         | 47.70         |
> | **TR-Certified (Ours)**  | 95.88     | 95.45     | 96.26    | 92.01     | 159.20    |
>
> We observe that when the target KL is below 0.10, our method’s runtime remains stable (around 92–96 seconds), and is reduced by nearly half compared to the case of KL = 0.10. These empirical results demonstrate that our method achieves a favorable utility–efficiency trade-off.
>
> Once again, we sincerely appreciate your detailed comments on our paper. Please share any further concerns with us.
>
> [1]Wang, Weiqi, et al. "Machine unlearning: A comprehensive survey." arXiv preprint arXiv:2405.07406 (2024).
>
> [2]Bourtoule, Lucas, et al. "Machine unlearning." 2021 IEEE symposium on security and privacy (SP). IEEE, 2021.
>
> [3]Mu, Siqiao, and Diego Klabjan. "Rewind-to-delete: Certified machine unlearning for nonconvex functions." arXiv preprint arXiv:2409.09778 (2024).

---

> ### Author Response · Authors · 2025-11-27
>
> Thank you again for your time and effort in reviewing our work.
>
> We hope you are enjoying the Thanksgiving holiday, and we hope you have had an opportunity to look over our responses. We appreciate your recognition and positive feedback on our work. We believe we have addressed the concerns raised about our paper, and we kindly hope you will consider updating your score accordingly to help provide a clear signal to the AC.
>
> As the discussion period will end in less than a week, we wish to leave enough time to address your futher concerns. If you have any further questions, we are happy to engage on addressing them.

---

### Official Review · Reviewer_7vHP · 2025-11-01

**Soundness:** 3
**Presentation:** 3
**Contribution:** 3
**Rating:** 6
**Confidence:** 3

**Summary:**

This paper introduces TR Certified Machine Unlearning, a trust region based framework that addresses a key limitation of certified unlearning methods based on Newton updates, namely the assumption that deleted samples are independent and identically distributed. The authors point out that in practical scenarios, deletion requests are often biased, leading to distribution shifts that undermine existing theoretical guarantees. TR Certified Machine Unlearning incorporates a trust region constraint into the Newton update process, adaptively controlling the update radius and step size to ensure that each iteration remains within a locally reliable region.

**Strengths:**

1. The paper establishes a clear mathematical link between biased deletion and distribution shift and shows why this invalidates conventional Newton-based unlearning. Theoretical derivations, including the local smoothness assumption, descent lemma, and convergence bounds, are carefully constructed and logically consistent, giving the work strong analytical credibility.

2. The experimental design closely mirrors the theoretical claims. The authors quantitatively relate posterior KL divergence to performance degradation and demonstrate that TR-Certified Unlearning effectively mitigates this effect. The alignment between theory and empirical outcomes makes the proposed framework both interpretable and verifiable.

3. TR-Certified Unlearning achieves smaller gradient residual bounds and thus requires less noise for certification while preserving model accuracy. This balance between formal privacy guarantees and practical model performance represents a meaningful improvement over existing certified unlearning approaches.

**Weaknesses:**

1. The experiments are based on small datasets such as MNIST and CIFAR-10, and the models used are relatively simple. It is uncertain whether the proposed method would still show a clear advantage when tested on larger and more complex networks. As model size grows, the extra computation and time required by the trust-region updates may reduce or even cancel out the observed gains.

2. It would be nice if the paper can include more baselines in the comparisons.

3. The paper briefly mentions potential applications but does not discuss how the proposed method would integrate into existing unlearning frameworks or data deletion pipelines. A short discussion on deployment practicality would make the contribution more complete.

**Questions:**

Please refer to the weaknesses.

---

> ### Author Response · Authors · 2025-11-21
> **Response to Reviewer 7vHP (Part 1 - 1/2)**
>
> We sincerely thank the reviewer for their careful reading and for the thoughtful effort to help improve our paper. We truly appreciate the recognition of our work’s conceptual novelty, algorithmic clarity, and evaluation design. In the following, we address the reviewer’s concerns regarding our paper.
>
> > **W1: The experiments are based on small datasets such as MNIST and CIFAR-10, and the models used are relatively simple. It is uncertain whether the proposed method would still show a clear advantage when tested on larger and more complex networks. As model size grows, the extra computation and time required by the trust-region updates may reduce or even cancel out the observed gains.**
>
> **A1:**  We appreciate the reviewer’s concern regarding scalability to larger models and datasets. Our method is evaluated on certain datasets and models to enable controlled comparison with existing certified unlearning baselines. In response to this concern, we have conducted additional experiments on SVHN dataset with ResNet18. Table 1 reports the $ \Delta \mathrm{F} _ 1 $, calcuated by $ \left| \Delta \mathrm{F} _ 1\left(w _ {\text{retrain}}\right) - \Delta \mathrm{F} _ 1\left(w _ {\text{unlearn}}\right) \right| $ on test datasets. We refer our evaluation detail and design to Section 6. of our paper. A smaller  $ \Delta \mathrm{F}_1 $ indicates better utility retention.
> To make the table concise, we only keep the $ \Delta \mathrm{F}_1 $. Table 2 demonstrates the time recorded in seconds on different datasets and model architecture during unlearning.
>
> **Table 1.** $ \Delta \mathrm{F}_1 $ across datasets and models at KL = 0.104.
> | Method                  | MNIST & MLP $ \Delta \mathrm{F}_1 $ | CIFAR-10 & AllCNN $ \Delta \mathrm{F}_1 $ | SVHN & ResNet18 $ \Delta \mathrm{F}_1 $ |
> |-------------------------|------------------|------------------------|---------------------|
> | Guo et al. (2020)       | 3.94               | NaN                    | NaN                  |
> | Zhang et al. (2024)     | 4.18             | 9.28                   | 11.20                 |
> | **TR-Certified (Ours)** | **1.37**         | **3.15**               | **4.09**                 |
>
> **Table 2.** Running time (wall-clock seconds) across datasets and models at KL = 0.104.
> | **Method**               | **MNIST & MLP (s)** | **CIFAR-10 & AllCNN (s)** | **SVHN & ResNet-18 (s)** |
> |--------------------------|---------------------|----------------------------|---------------------------|
> | Guo et al. (2020)        | 28.80               | NaN                        | NaN
> | Zhang et al. (2024)      | 14.03               | 47.7                       | 105.13
> | **TR-Certified (Ours)**  | 48.23               | 158.2                      | 354.72
>
> As shown in the first utility table, our method consistently outperforms the baselines by approximately 3× on the evaluated metrics. As a tradeoff, we observe in Table 2 that the our method shows longer unlearning time than other baselines. Noted that we observe a uniform runtime increase across datasets and architectures, suggesting that our method scales predictably with model size and data volume. This indicates that our proposed method’s performance advantages remain stable on larger datasets and more complex models.
>
> > **W2: It would be nice if the paper can include more baselines in the comparisons.**
>
> **A2:**  We appreciate the reviewer's insight to provide more baselines[3]. In response to your concern, we have conducted an additional experiment on another baseline that also utilizes Newton updates for certified unlearning. We are currently working on incorporating more relevant baselines into our evaluation and will update the paper accordingly. Referring to the table:
>
> **Table 1.** $ \Delta \mathrm{F}_1 $ across datasets and models at KL = 0.104.
> | Method                  | MNIST & MLP $ \Delta \mathrm{F}_1 $ | CIFAR-10 & AllCNN $ \Delta \mathrm{F}_1 $ | SVHN & ResNet18 $ \Delta \mathrm{F}_1 $ |
> |-------------------------|------------------|------------------------|---------------------|
> | Guo et al. (2020)       | 3.94              | NaN                    | NaN                  |
> | Zhang et al. (2024)     | 4.18             | 9.28                   | 11.20                 |
> | **TR-Certified (Ours)** | **1.37**         | **3.15**               | **4.09**                 |
>
> It is clear that our method remains top performances across different baselins.

---

> ### Author Response · Authors · 2025-11-21
> **Response to Reviewer 7vHP (Part 2 - 2/2)**
>
> > **W3: The paper briefly mentions potential applications but does not discuss how the proposed method would integrate into existing unlearning frameworks or data deletion pipelines. A short discussion on deployment practicality would make the contribution more complete.**
>
> **A3:** Our method can be integrated into unlearning pipeline with white-box access as post-unlearning correction step[2]. For example, in a SISA-style pipeline[1] for a retail image classifier, after deleting a user’s data and partially retraining the affected shards, we can replace the old post step with one trust-region correction. Beyond certified unlearning, our proposed method can also merge into general post-unlearning pipeline by skipping the certification step given white-box access.
>
> As an important future direction, we plan to investigate how to best integrate our method into existing unlearning frameworks. We appreciate the reviewer for highlighting this valuable question.
>
> Once again, we sincerely appreciate your detailed comments on our paper. Please share any further concerns with us. We will keep updating other baselines.
>
> [1]Bourtoule, Lucas, et al. "Machine unlearning." 2021 IEEE symposium on security and privacy (SP). IEEE, 2021.
>
> [2]Mu, Siqiao, and Diego Klabjan. "Rewind-to-delete: Certified machine unlearning for nonconvex functions." arXiv preprint arXiv:2409.09778 (2024).
>
> [3]Guo, Chuan, et al. "Certified data removal from machine learning models." arXiv preprint arXiv:1911.03030 (2019).

---

> ### Author Response · Authors · 2025-11-27
>
> Thank you again for your time and effort in reviewing our work.
>
> We hope you are enjoying the Thanksgiving holiday, and we hope you have had an opportunity to look over our responses. We appreciate your recognition and positive feedback on our work. We believe we have addressed the concerns raised about our paper, and we kindly hope you will consider updating your score accordingly to help provide a clear signal to the AC.
>
> As the discussion period will end in less than a week, we wish to leave enough time to address your futher concerns. If you have any further questions, we are happy to engage on addressing them.

---

### Official Review · Reviewer_znGP · 2025-11-01

**Soundness:** 2
**Presentation:** 3
**Contribution:** 3
**Rating:** 2
**Confidence:** 4

**Summary:**

The paper addresses certified unlearning under biased deletions that induce distribution shift. Instead of a single damped-Newton step, it wraps approximate Newton updates in a trust-region loop: build a local quadratic model, cap the step by a clipped radius, accept or reject using a model-agreement ratio, and iterate. The theory provides a pre-run gradient bound based on local Lipschitz constants estimated in the current trust ball, and the implementation relies on HVPs with truncated CG/LiSSA. The claim is that this yields tighter certificates and tracks the retrained solution better than a one-step Newton method when the deletion set is non-i.i.d.

**Strengths:**

1.	The trust-region wrapper around an approximate Newton step is technically coherent: it preserves a Newton-like search direction while controlling step size by a model-agreement rule under a locally defined ball.

2.	The certificate is tied to quantities the algorithm actually estimates in the loop, namely a local gradient Lipschitz constant in the current ball and a spectral-norm bound from power iteration, which makes the residual bound operational rather than symbolic.

**Weaknesses:**

1. The clipping rule sets r_t=∥g_t∥/L_t, while L_titself is defined as the local gradient-Lipschitz constant on the ball B(w_t,r_t). Without an a priori radius or an update scheme that defines L_tbefore r_t, the pair (r_t,L_t)is not well defined and the step size cannot be computed from the specification as written.

2. The paper asserts that, under distribution shift, the trust-region scheme “achieves a closer approximation to the retrained model than a one-step Newton update,” yet the formal results shown in the same section are a pre-run gradient bound and a descent control based on L_max (T). There is no explicit inequality or counterexample in the same metric that compares the trust-region iterate to the one-step Newton solution, so the central comparison is stated but not actually established.
3. To secure positive definiteness of the approximate Hessian, the analysis assumes a damping shift by λI. This is equivalent to inserting strong l_2regularization on the retained objective. The regularization level simultaneously makes the theory go through and changes the very objective being certified; the text acknowledges the damping and its analogy to l_2 but does not separate the effect on certifiability from the change in the target function.

**Questions:**

Please refer to the weakness above.

---

> ### Author Response · Authors · 2025-11-21
> **Response to Reviewer znGP (Part 1)**
>
> We are thankful for the reviewer’s interest in our study and the helpful suggestion. We appreciate the reviewer’s feedback on our paper. In the following, we address the reviewer’s concerns regarding our paper and respond to the reviewer’s questions.
>
> > W1: The clipping rule sets r _ t=∥g _ t∥/L _ t, while L_titself is defined as the local gradient-Lipschitz constant on the ball B(w_t,r_t).
>
> **A1:** We appreciate the reviewer’s observation. In our implementation, the pair is well defined by first computing an independent local Lipschitz upper bound $ L_t $ at $ w_t $ , and only then setting the radius.
>
> Specifically, we estimate the initial $ \widehat{L}_t $ via a few Hessian-vector product power iterations. This leads to  $ r_t = \Vert g_t \Vert / \widehat{L}_t $ . This makes $ (r_t, \widehat{L}_t) $ well defined at every iteration and preserves our certificates by using $ L _ {\max}(T) = \max _ {t \le T} \widehat{L} _ t $ . In implementation, we avoid calculating the $\widehat L_t$ every time, so that we apply an iterative update which is defined as:
> $
> \widehat L _ t = \alpha _ L\,\widehat L _ {t-1}, \qquad \alpha _ L \ge 1,
> $
>
> to quickly get the $\widehat L_t$ for next step. We omit this part in the paper as this only a implementation technique to use. In respond, we will add this clarification in appendix.
>
> We then follow the radius update in each step described in appendix C. No theoretical results change need, but we will add this clarification to avoid ambiguity.
>
>
> > W2: There is no explicit inequality or counterexample in the same metric.
>
> **A2:**  In the paper we give plain text explanation that our method achieve closer approximation under Definition 3. We appreciate the your insight regarding this point and we will add this additional clarification theorem in to our revised paper.
>
> We measure ``closeness'' to the retrained model using the residual gradient of the retained objective, $\|\nabla \ell_R(\cdot)\|$. Specifically, under the standard local curvature conditions we already use($\mu$--strong convexity with $L$--smoothness), we derive residual gradient controls distance:
>
> $$
> \|w_t - w^\star\| \\le\ \tfrac{1}{\mu}\|\nabla \ell_R(w_t)\|
> $$
>
> Consequently, bounding $\|\nabla \ell_R\|$ rigorously bounds closeness to the retrained model. We give a formal mathematical inequality by directly compares the worst-case certified upper bound on the residual gradient of the retained objective. Under the local assumptions where $\mu$-strong convexity with $L$-smoothness, one step of our trust-region (TR) update satisfies:
>
> $$
> \|\nabla \ell_R(w_t+p_{\mathrm{TR}})\| \le \kappa_{\mathrm{TR}}\|g_t\| +\ \frac{M}{2}r_t^2,
> \quad\text{where }\
> \kappa_{\mathrm{TR}} = \frac{\lambda}{\mu+\lambda}
> $$
>
> For a one-step Newton update with a approximated, mis-specified curvature $H_{\mathrm{est}}$, we have:
>
> $$
> \|\nabla \ell_R(w_t+p_N)\| \le \alpha_N \|g_t\| +\ \frac{M}{2}\|H_{\mathrm{est}}^{-1}\|^2\|g_t\|^2,
> \quad\text{where }\
> \alpha_N = \|I - H_t H_{\mathrm{est}}^{-1}\|
> $$
>
> Given this inequality, we ensures a closer approximation by comparing the upper bound of these two method and we (i) keeps $ r _ t \le \|(H_{\mathrm{est}}+\lambda_N I)^{-1}\|\|g_t\|$ , and (ii) keep $\kappa_{\mathrm{TR}} \le \|I - H_t(H_{\mathrm{est}}+\lambda_N I)^{-1}\|$. This ensures our certified bound is no larger than the single Newton update bound, indicating a closer approximation.

---

> ### Author Response · Authors · 2025-11-21
> **Response to Reviewer znGP (Part 2)**
>
> > W3:  To secure positive definiteness of the approximate Hessian, the analysis assumes a damping shift by λI. This is equivalent to inserting strong l_2regularization on the retained objective. The regularization level simultaneously makes the theory go through and changes the very objective being certified; the text acknowledges the damping and its analogy to l_2 but does not separate the effect on certifiability from the change in the target function.
>
> **A3:** Thank you for raising this point. In our paper, our certificates are with respect to $\lambda $. We add the damping only to ensure a local curvature and stability. This implies certification with respect to the $\lambda$-regularized objective, but the gap to the unregularized target is small and bounded under our local curvature assumptions. Please refer to previous papers[1][2][3] for the bound. Consequently, the effect on certifiably from the target change does not alter our evaluation protocol or the main conclusions.
>
> In addition, damping by adding $+\lambda I$ is a standard technique to ensure a positive-definite curvature model in second-order/trust-region methods[1][2][3] and is not the central contribution of our work. We appreciate the reviewer’s careful attention here.
>
> Once again, we sincerely appreciate the reviewer's detailed comments on our paper. We believe that we have addressed the reviewer's main concerns about the paper. Please share any further concerns with us.
>
> [1]Sekhari, Ayush, et al. "Remember what you want to forget: Algorithms for machine unlearning." Advances in Neural Information Processing Systems 34 (2021): 18075-18086.
>
> [2]Suriyakumar, Vinith, and Ashia C. Wilson. "Algorithms that approximate data removal: New results and limitations." Advances in Neural Information Processing Systems 35 (2022): 18892-18903.
>
> [3]Zhang, Binchi, et al. "Towards certified unlearning for deep neural networks." arXiv preprint arXiv:2408.00920 (2024).

---

> ### Author Response · Authors · 2025-11-27
>
> Thank you again for your time and effort in reviewing our work.
>
> We hope you are enjoying the Thanksgiving holiday, and we hope the reviewer has had an opportunity to look over our responses. We believe we have addressed the concerns raised about our paper, and we kindly hope the reviewer will consider updating their score accordingly.
>
> As the discussion period will end in less than a week, we wish to leave enough time to address your futher concerns. If you have any further questions, we are happy to engage on addressing them.

---

### Author Response · Authors · 2025-12-03
**Summary of Discussions**

Dear Area Chair and reviewers,

We are grateful for your time in assessing our submission. Here we provide a brief summary of our paper (including its contribution) and our responses to all the reviews.

Our paper is, to the best of our knowledge, the **first** to identify that distribution shift degrades the performance of certified unlearning, and to propose a novel certified unlearning framework that remains effective and efficient under distribution shift by using iterative Newton updates constrained by a trust region. As a comparison, all prior certified unlearning works (including those we cite and those mentioned by the reviewers) assume i.i.d. unlearned and retained data—they may unlearn non-i.i.d. data, but the unlearned model’s performance degrades significantly.

**Many reviewers recognized the strengths of our work, including its theoretical soundness (7vHP, fafB), novel problem statement (fafB), experimental design (7vHP), and practical importance (t1Q3). Additionally, we summarize how we addressed three main concerns during the rebuttal stage below.**

---

### Comment (1)[ znGP, fafB] **Theoretical comparison between our approach and prior work (one-step Newton update).**

**Response and our solution:**  Our original description was in plain text under Definition 3.  As a response, we provide an explicit, same-metric comparison formula in “Response to Reviewer znGP (Part 1)” and formalize it as Theorem 3 in Appendix A.

We also pointed out that Appendix B (Proposition 4, Taylor remainder under optimizer shift) systematically analyzes the mathematical reasons behind the failure of previous single-step Hessian updates on non-i.i.d. forget sets in “Response to Reviewer fafB (Part 2)”.

---

### Comment (2) [fafB, 7vHP, t1Q3] **Empirical comparison with other baselines with more complex, modern architectures and larger datasets.**

**Response and our solution:** We added additional empirical comparison in “Response to Reviewer fafB (Part 1)”, “Response to Reviewer 7vHP (Part 1 - 1/2)”, and in Table 1 of the revised manuscript.  Our results show that our approach outperforms both baselines in terms of delta F1 score between the unlearned model and the retrained model.

We also added a comparison with prior works in terms of runtime overhead in “Response to Reviewer t1Q3 (Part 2 -2/2)”, and in Appendix D. We acknowledge that there exists a trade-off between runtime overhead and unlearning performance. At the cost of running slower than the baselines, our method still remains faster than full retraining and achieves substantially higher F1, yielding a more favorable utility-efficiency balance.


---

### Comment (3) [t1Q3] **Research novelty**
**Response and our solution:** We emphasized that the main contribution of our paper is to derive a tight upper bound on the residual gradient within the trust-region framework, which goes well beyond a simple combination of existing ideas. We clarified this in Response to Reviewer t1Q3 (Part 1 -1/2), and emphasized our novelty in introduction.

---

Final Notes: We also responded to all other minor clarification points (many of which were already covered in our original appendix and explicitly referenced in our responses) and skip the details here for brevity. We would be grateful if the AC could refer to the detailed responses if interested. During the interactive rebuttal stage, unfortunately, none of the reviewers was able to participate, especially given that the rebuttal period was significantly cut due to unforeseen reasons. We sincerely hope the AC will consider our thorough response when making decisions.

Authors

---

### Meta-Review · Area_Chair_jtYb · 2026-01-07

**Summary:**

1. The paper's central theoretical claim, i.e., superiority over a Newton update, is stated but not formally proven.
2. The method's practical application is unclear or limited.
3. The experiments are limited to toy datasets such as MNIST and CIFAR10.
4. The method's significant computational overhead prevents scaling to large models, which limits the scope of this work.

**Reviewer Concerns:**

Points 1 and 3 were partially addressed, but points 2 and 4 remain outstanding.

**Reviewer Scores:**

No change

---

### Decision · Program_Chairs · 2026-01-26

Reject